# The genetic risk of gestational diabetes in South Asian women

Amel Lamri[1,2], Jayneel Limbachia[2,3], Karleen M Schulze[2], Dipika Desai[2], Brian Kelly[4], Russell J de Souza[2,3], Guillaume Paré[2,3,5], Deborah A Lawlor[6,7,8], John Wright[4], Sonia S Anand[1,2,3]*, On behalf of for the Born in Bradford and START investigators

[1]Department of Medicine, McMaster University, Hamilton, Canada; [2]Population Health Research Institute, Hamilton, Canada; [3]Department of Health Research Methods, Evidence, and Impact, McMaster University, Hamilton, Canada; [4]Bradford Institute for Health Research, Bradford Royal Infirmary, Bradford, United Kingdom; [5]Department of Pathology and Molecular Medicine, McMaster University, Hamilton, Canada; [6]Population Health Science, Bristol Medical School, University of Bristol, Bristol, United Kingdom; [7]MRC Integrative Epidemiology Unit, University of Bristol, Bristol, United Kingdom; [8]Bristol NIHR Biomedical Research Centre, Bristol, United Kingdom

*For correspondence:
anands@mcmaster.ca

**Abstract** South Asian women are at increased risk of developing gestational diabetes mellitus (GDM). Few studies have investigated the genetic contributions to GDM risk. We investigated the association of a type 2 diabetes (T2D) polygenic risk score (PRS), on its own, and with GDM risk factors, on GDM-related traits using data from two birth cohorts in which South Asian women were enrolled during pregnancy. 837 and 4372 pregnant South Asian women from the SouTh Asian BiRth CohorT (START) and Born in Bradford (BiB) cohort studies underwent a 75-g glucose tolerance test. PRSs were derived using genome-wide association study results from an independent multi-ethnic study (~18% South Asians). Associations with fasting plasma glucose (FPG); 2 hr post-load glucose (2hG); area under the curve glucose; and GDM were tested using linear and logistic regressions. The population attributable fraction (PAF) of the PRS was calculated. Every 1 SD increase in the PRS was associated with a 0.085 mmol/L increase in FPG ([95% confidence interval, CI=0.07–0.10], p=2.85×10$^{-20}$); 0.21 mmol/L increase in 2hG ([95% CI=0.16–0.26], p=5.49×10$^{-16}$); and a 45% increase in the risk of GDM ([95% CI=32–60%], p=2.27×10$^{-14}$), independent of parental history of diabetes and other GDM risk factors. PRS tertile 3 accounted for 12.5% of the population's GDM alone, and 21.7% when combined with family history. A few weak PRS and GDM risk factors interactions modulating FPG and GDM were observed. Taken together, these results show that a T2D PRS and family history of diabetes are strongly and independently associated with multiple GDM-related traits in women of South Asian descent, an effect that could be modulated by other environmental factors.

## Editor's evaluation

South Asian women have twice the risk of developing Gestational Diabetes Mellitus (GDM) compared with white European women. This clearly presented comprehensive study shows that a T2D polygenic risk score is strongly associated with multiple GDM-related traits in South Asian women and is a significant contributor to the population-attributable fraction of GDM, independently of family history of diabetes. This will be of interest to genetic epidemiologists and clinicians working in this field.

## Introduction

Gestational diabetes mellitus (GDM) is defined as hyperglycemia first diagnosed during pregnancy. This abnormal increase in blood glucose levels is associated with an increased risk of adverse health outcomes for both mother and their fetus/child during pregnancy, and later in life (*Farrar et al., 2016*). It is estimated that 1% to >30% of live births are affected by GDM worldwide. This prevalence has been shown to vary widely depending on the participants ethnicity, countries/regions, and on the diagnostic criteria used (*Archambault et al., 2014*; *McIntyre et al., 2019*). South Asian women (whose ancestry derives from the Indian subcontinent) have a twofold increased odds of developing GDM, compared to white European women (*Anand et al., 2016*; *Cosson et al., 2014*; *Farrar et al., 2015*; *McIntyre et al., 2019*). The reasons for this disproportionate risk have not been fully characterized.

Gestational diabetes is a complex disorder influenced by multiple genetic and environmental factors such as maternal age, ethnicity, obesity, poor diet quality, and family history of diabetes (*Anand et al., 2017*; *Hedderson et al., 2011*; *Solomon et al., 1997*). Most genetic and environmental GDM risk factors are shared with type 2 diabetes (T2D; *Sattar and Greer, 2002*; *Zhang and Ning, 2011*) another condition that is thought to be very closely related to GDM. For example, women with GDM have a higher probability of having at least one parent with T2D, compared to those with normal gestational glycemia (*Jang et al., 1998*). Furthermore, women with a GDM history have a tenfold higher risk of subsequently being diagnosed with T2D compared to those without a history of GDM (*Vounzoulaki et al., 2020*). In terms of genetic architecture, both candidate gene and genome-wide association studies (GWASs) demonstrated a considerable overlap between GDM and T2D (*Hayes et al., 2013*; *Kwak et al., 2012*; *Pervjakova et al., 2022*). Finally, T2D polygenic risk scores (PRSs) have also been associated with GDM risk (*Lamri et al., 2020*; *Pervjakova et al., 2022*).

It has been demonstrated that environmental exposures such as diet and/or physical activity may modulate the effect of T2D loci (such as *TCF7L2, PPARG,* and *CDKAL1*) on the risk of T2D (*Dietrich et al., 2019*). Nevertheless, only a handful of studies have investigated genetic×environmental interactions on GDM (*Chen et al., 2019*; *Grotenfelt et al., 2016*; *Popova et al., 2017*), and to date, no study has tested the interaction between a genome-wide PRS with other GDM risk factors, on the risk of GDM.

The aims of this investigation were to: (i) test the association of a T2D PRS, generated from an external multi-ethnic GWAS (~18% South Asians), with GDM and related traits (fasting plasma glucose [FPG], 2 hr post-load glucose (2hG), and area under the curve glucose [AUCg] levels) in pregnant South Asian women from the SouTh Asian biRth cohorT (START) and the Born in Bradford (BiB) studies; (ii) To estimate the population attributable fraction (PAF) of the PRS on GDM; and (iii) To determine whether the effect of the PRS is modulated by other GDM risk factors including age, BMI, diet quality, birth country, education, and parity.

## Results

The proportion of women classified with GDM using the IADPSG criteria was 25% and 11.2% in START and BiB, respectively, which was lower than the proportion using the South Asian-specific definition of 36.2% and 22.9%, respectively. Notably the proportion of women with GDM was higher in START compared to BiB irrespective of the classification method used.

The proportion of women of Indian origin in START and BiB was 71.8% and 5.1%, while the proportion of Pakistani women was 23.4% and 94.3%, respectively. The proportion of participants born in the Indian sub-continent was higher in START (88.6%) than in BiB (55.6%), and the average number of years spent in Canada or the United Kingdom among these participants was lower in START compared to BiB (6.6 vs. 9.7 years, respectively). The proportions of primiparous women (40.9% vs. 31.7%) and women with one prior pregnancy (42.4% vs.26.9%) were higher in START than in BiB. Conversely, participants with two or more prior pregnancies were more frequent in BiB than START (41.4% vs. 16.6%, respectively). The proportion of vegetarian participants was higher in START than in BiB (36.4% vs. 1.3%). Finally, the proportion of participants with a post-secondary degree/diploma or higher was greater in START than BiB (84.0% vs. 29.0%).

The standardized PRS ranged between –3.23 and 3.12 in START as compared to –3.51 and 4.16 in BiB. The full list of genetic variants included in the PRS as well as their characteristics are shown in *Supplementary file 1a*.

**Table 1.** Characteristics of START and BiB study participants included in the analysis.

| | START | | | BiB | | |
|---|---|---|---|---|---|---|
| | **No GDM** | **GDM** | **p Value** | **No GDM** | **GDM** | **p Value** |
| N (%) | 759 (75) | 253 (25) | – | 3809 (88.8) | 481 (11.2) | – |
| Age, years | 29.8 (3.8) | 31.6 (4) | $5.55×10^{-10}$ | 27.7 (5) | 30.5 (5.4) | $1.40×10^{-22}$ |
| Height, cm | 162.5 (6.27) | 161.13 (6.01) | 0.002 | 159.9 (5.69) | 158.3 (5.66) | $6.19×10^{-08}$ |
| Weight, kg [a] | 61.7 (11.7) | 65.6 (12.9) | $2.00×10^{-05}$ | 64.8 (14.1) | 71.1 (15.1) | $2.22×10^{-16}$ |
| BMI, kg/m² [b] | 23.4 (4.3) | 25.3 (4.9) | $4.93×10^{-08}$ | 25.4 (5.2) | 28.4 (5.8) | $5.89×10^{-23}$ |
| Parity, n (%) | | | | | | |
| 0 | 328 (44.3%) | 78 (31.1%) | | 1189 (32.2%) | 129 (27.4%) | |
| 1 | 299 (40.4%) | 122 (48.6%) | 0.001 | 1026 (27.8%) | 94 (20%) | $5.03×10^{-07}$ |
| 2 or more | 114 (15.4%) | 51 (20.3%) | | 1473 (39.9%) | 248 (52.7%) | |
| Post-secondary education, n (%) | 641 (84.6%) | 208 (82.2%) | 0.43 | 952 (29.4%) | 110 (26.3%) | 0.22 |
| Country of origin/ancestry, n (%) | | | | | | |
| India | 567 (74.7%) | 160 (63.2%) | | 175 (5.2%) | 19 (4.3%) | |
| Pakistan | 163 (21.5%) | 74 (29.2%) | 0.001 | 3198 (94.2%) | 425 (95.5%) | 0.37 [c] |
| Other | 29 (3.8%) | 19 (7.5%) | | 23 (0.7%) | 1 (0.2%) | |
| Born in South Asia, n (%) | 671 (88.5%) | 225 (88.9%) | 0.95 | 1836 (54.2%) | 291 (65.7%) | $6.50×10^{-06}$ |
| Years in recruitment country (Canada/UK) [d] | 6.4 (5.8) | 7.4 (5.8) | 0.02 | 9.3 (9) | 12.1 (9.4) | $3.88×10^{-06}$ |
| Parental history of diabetes, n (%) | 282 (37.3%) | 142 (56.1%) | $2.25×10^{-07}$ | 891 (27.4%) | 170 (38.9%) | $8.88×10^{-07}$ |
| Vegetarians, n (%) | 266 (37%) | 84 (34.6%) | 0.54 | 12 (1.3%) | 1 (1.1%) | >0.99 [c] |
| Low diet quality, n (%) | 180 (24) | 88 (35.1) | $8.00×10^{-04}$ | – | – | – |
| Polygenic risk score (z-scores) | –0.11 (1) | 0.347 (0.93) | $1.51×10^{-08}$ | –0.04 (0.99) | 0.32 (1.04) | $4.98×10^{-12}$ |
| Polygenic risk score | | | | | | |
| Tertile 1 | 240 (37.7%) | 39 (19.4%) | | 1309 (34.4%) | 117 (24.3%) | |
| Tertile 2 | 206 (32.4%) | 73 (36.3%) | $2.74×10^{-06}$ | 1291 (33.9%) | 142 (29.5%) | $7.60×10^{-10}$ |
| Tertile 3 | 190 (29.9%) | 89 (44.3%) | | 1209 (31.7%) | 222 (46.2%) | |
| Fasting plasma glucose, mmol/L | 4.27 (0.32) | 5.02 (0.83) | $5.51×10^{-32}$ | 4.53 (0.41) | 5.34 (1.14) | $3.18×10^{-43}$ |
| 1 hr post-load glucose, mmol/L | 7.31 (1.38) | 10.26 (2.02) | $6.04×10^{-57}$ | – | – | – |
| 2 hr post-load glucose, mmol/L | 5.96 (1.16) | 8.47 (2.16) | $1.53×10^{-42}$ | 5.49 (1.02) | 9.14 (1.97) | $1.57×10^{-155}$ |
| Area under curve glucose, mmol.hr [e] | 12.43 (1.83) | 17.02 (2.89) | $2.27×10^{-63}$ | 10.02 (1.21) | 14.48 (2.77) | $3.82×10^{-133}$ |

Characteristics of participants with available PRS and GDM IADPSG, FPG, 1 hr, 2 hr post-load glucose levels or AUC glucose data. Presented data are means (standard deviation) unless otherwise indicated. p Values are calculated from Chi-squared test for categorical variables and independent t-test for continuous variables. [a] Pre-pregnancy values in START vs. weight at antenatal clinic (average 12 completed weeks of pregnancy) in BiB. [b] Derived using height measured at initial visit (in both studies) and pre-pregnancy weights (START) or antenatal clinic weights (BiB). [c] Approximation may be incorrect due to small counts. [d] Canada for START samples and UK for BiB. [e] Derived using fasting, 1 hr and 2 hr post-load measurements in START vs. fasting and 2 hr post-load measurements in BiB. Abbreviations: AUC, area under the curve glucose; BiB, Born in Bradford; BMI, body mass index; GDM, gestational diabetes mellitus; IADPSG, International Association of Diabetes and Pregnancy Study Groups; START, south Asian birth cohort; T2D, type 2 diabetes; UK, United Kingdom; vs., versus.

*Table 1* shows the baseline characteristics of the South Asian women from the START and BiB stratified by GDM case versus non GDM (IADPSG criteria). As expected, women with GDM had a higher mean fasting, 2hG and AUCg levels than non-GDM participants. Participants with GDM were older, had a higher BMI, and were more likely to report a family history of diabetes compared to women without GDM, in both studies. The overall diet quality was lower in participants with GDM compared to non-GDM participants in START (data not available in BiB). Of note, the average difference in BMI

between GDM cases and controls was higher in BiB than in START (3.0 and 1.9, respectively) (*Table 1*). Women with GDM had a higher mean PRS compared to women without GDM. Similarly, women with GDM were more likely to have PRS categorized in tertile 2 or 3, compared to tertile 1 (*Table 1*).

## Genetic risk and GDM-related traits in univariate models

The continuous PRS was associated with FPG, 2hG, and AUCg in START and BiB in univariate models. Every 1 SD increase in the PRS was associated with a 0.09 mmol/L increase in FPG (95% confidence interval [CI]=0.07–0.10), 0.23 mmol/L increase in 2hG (95% CI=0.18–0.28), and a 0.17 unit increase in AUCg z-scores (0.14–0.20) in the meta-analyzed results (*Supplementary file 1b*).

The PRS was also associated with the risk of $GDM_{IADPSG}$ in univariate models whereby a 1 SD increase in PRS was associated with a 47% increase in risk of GDM after meta-analysis (95% CI=35–60%). A similar association is observed using the South Asian-specific definition of GDM, with moderate between-study heterogeneity observed (*Supplementary file 1b*).

Overall, the risk of $GDM_{IADPSG}$ increased progressively comparing tertile 2 of the PRS to tertile 1, and tertile 3 to tertile 1 (43% and 230%, respectively; *Supplementary file 1b*). Higher PRS categories were also associated with higher FPG, 2hG, and AUCg levels (*Supplementary file 1b*).

## Multivariable models of GDM risk factors and GDM-related traits

The continuous PRS was strongly and independently associated with FPG, 2hG, and AUCg levels in a multivariable model adjusted for age, BMI, parity, parental history of diabetes, region of birth (South Asia vs. other), education level, and diet quality (available in START only), and the first five PCs (*Table 2*). For example, every 1 SD increase in the PRS was associated with a 0.08 mmol/L increase in FPG, and 0.21 mmol/L increase in 2hG levels (*Table 2*). The continuous PRS was also associated with a higher risk of GDM in a model with similar adjustments, whereby every 1 SD increase in the PRS was associated with a 45% increase in the risk of $GDM_{IADPSG}$ (*Table 2*). Similar association results for GDM using the South Asian-specific criteria were observed and are shown in *Supplementary file 1c*.

When testing tertiles of PRS with similar covariates, our results show that participants in the second and third PRS tertiles have a 37% and 119% increase in the risk of $GDM_{IADPSG}$ compared to participants in tertile 1, respectively (*Supplementary file 1d*). Higher PRS tertiles were also associated with higher FPG, 2hG, and AUCg levels (*Supplementary file 1d*). The effect sizes associated with tertiles 2 were higher in START than BiB across multiple GDM-related traits (2hG, AUCg, and GDM; *Supplementary file 1d*).

## Population attributable fraction and detection rate

In a model adjusted for maternal age, BMI, education, birth in South Asia (yes/no), parental history of diabetes, and diet quality (in START only), the PRS tertile 3 accounted for 12.5% of the population's total $GDM_{IADPSG}$ cases overall, and was higher in START than in BiB (*Table 3*). The combined effect of PRS and parental history of diabetes on GDM accounted for ~21.7% of the population's GDM cases in the two studies combined (*Table 3*).

The detection rate associated with the top versus lower PRS tertile was equal to 10% for a 5% false positive rate.

## Interactions between the PRS and GDM risk factors on GDM

No consistent interactions were observed between the PRS and maternal age; parity; or education level modulating FPG, 2hG, AUCg, or GDM in START or BiB (*Table 4* and *Supplementary file 1e*).

A couple of nominally significant interactions modulating the continuous trait of FPG were observed in START were not confirmed in BiB and vice versa. These included the PRS×BMI and the PRS×birth in South Asia (yes/no) interactions (START $P_{interaction}$=0.01 and 0.04, respectively), yet non-significant in BiB ($P_{interaction\ PRS×BMI}$=0.05 and $P_{interaction\ PRS×birth\ in\ South\ Asia}$=0.07), with different effect sizes and opposing direction of effect between the two studies (*Supplementary file 1f*), resulting in non-significant meta-analysis of these effects ($P_{interaction\ PRS×BMI}$=0.42 and $P_{interaction\ PRS×birth\ in\ South\ Asia}$=0.26, respectively). Another interaction between the PRS and BMI modulating the risk of GDM was observed in BiB ($P_{interaction}$=0.03), but not in START ($P_{interaction}$=0.15; *Table 4*). Given that the overall direction of effect was similar in the two studies, this interaction remained significant after meta-analysis ($P_{interaction}$=0.01). Nevertheless, this result in START could be a false negative given the study's smaller sample size (with a power to

**Table 2.** Association between GDM risk factors and GDM-related traits: results from multivariate models in START and BiB cohorts.

| Dependent variable | Independent variables | START | | BiB | | Meta-analysis | | | |
|---|---|---|---|---|---|---|---|---|---|
| | | Beta/OR [95% CI] | p Value | Beta/OR [95% CI] | p Value | Beta (SE)/OR [95% CI] | p Value | I² | Q_E p value |
| Fasting glucose | PRS (per 1 SD increase) | 0.083 [0.043–0.123] | 6.00×10⁻⁰⁵ | 0.085 [0.065–0.105] | 1.67×10⁻¹⁶ | 0.085 [0.067–0.103] | 2.85×10⁻²⁰ | 0 | 0.92 |
| | Age (year) | 0.021 [0.01–0.032] | 2.00×10⁻⁰⁴ | 0.014 [0.009–0.019] | 2.49×10⁻⁰⁸ | 0.015 [0.011–0.02] | 4.19×10⁻¹¹ | 22 | 0.26 |
| | BMI (kg/m²) | 0.024 [0.014–0.033] | 5.51×10⁻⁰⁷ | 0.032 [0.028–0.036] | 6.53×10⁻⁵³ | 0.031 [0.027–0.034] | 7.99×10⁻⁶⁰ | 63 | 0.1 |
| | Born in South Asia (Yes/No) | 0.037 [−0.088 to 0.162] | 0.56 | 0.08 [0.039–0.122] | 2.00×10⁻⁰⁴ | 0.076 [0.037–0.115] | 2.00×10⁻⁰⁴ | 0 | 0.52 |
| | Parental history of T2D (Yes/No) | 0.04 [−0.043 to 0.123] | 0.34 | 0.066 [0.02–0.111] | 0.005 | 0.06 [0.02–0.1] | 0.003 | 0 | 0.6 |
| | Parity | −0.046 [−0.102 to 0.01] | 0.11 | −0.015 [−0.033 to 0.004] | 0.13 | −0.018 [−0.036 to 0] | 0.05 | 9 | 0.29 |
| | Education level (per level) | −0.031 [−0.068 to 0.006] | 0.1 | −0.016 [−0.035 to 0.002] | 0.09 | −0.019 [−0.036 to −0.002] | 0.02 | 0 | 0.49 |
| | Low diet quality (Yes/No) | 0.102 [0.01–0.193] | 0.03 | – | – | – | – | – | – |
| 2 hr postload glucose | PRS (per 1 SD increase) | 0.189 [0.068–0.311] | 0.002 | 0.211 [0.156–0.266] | 7.87×10⁻¹⁴ | 0.207 [0.157–0.257] | 5.49×10⁻¹⁶ | 0 | 0.75 |
| | Age (year) | 0.127 [0.093–0.161] | 8.34×10⁻¹³ | 0.068 [0.055–0.082] | 8.82×10⁻²³ | 0.076 [0.064–0.089] | 1.49×10⁻³² | 90 | 0.002 |
| | BMI (kg/m²) | 0.047 [0.019–0.074] | 0.001 | 0.064 [0.053–0.075] | 1.43×10⁻²⁹ | 0.062 [0.051–0.072] | 3.44×10⁻³² | 23 | 0.25 |
| | Born in South Asia (Yes/No) | 0.298 [−0.08 to 0.675] | 0.12 | 0.308 [0.195–0.422] | 1.14×10⁻⁰⁷ | 0.308 [0.199–0.416] | 3.09×10⁻⁰⁸ | 0 | 0.96 |
| | Parental history of T2D (Yes/No) | 0.361 [0.109–0.613] | 0.005 | 0.242 [0.117–0.366] | 1.00×10⁻⁰⁴ | 0.265 [0.154–0.377] | 3.23×10⁻⁰⁶ | 0 | 0.4 |
| | Parity | −0.279 [−0.45 to −0.109] | 0.001 | −0.095 [−0.146 to −0.043] | 3.00×10⁻⁰⁴ | −0.11 [−0.16 to −0.061] | 1.00×10⁻⁰⁵ | 76 | 0.04 |
| | Education level (per level) | −0.063 [−0.176 to 0.051] | 0.28 | −0.073 [−0.124 to −0.022] | 0.005 | −0.071 [−0.118 to −0.025] | 0.002 | 0 | 0.87 |
| | Low diet quality (Yes/No) | 0.365 [0.086–0.644] | 0.01 | – | – | – | – | – | – |
| AUC glucose | PRS (per 1 SD increase) | 0.165 [0.099–0.231] | 1.08×10⁻⁰⁶ | 0.152 [0.119–0.185] | 2.50×10⁻¹⁹ | 0.155 [0.125–0.184] | 7.74×10⁻²⁵ | 0 | 0.74 |
| | Age (per year) | 0.068 [0.05–0.087] | 1.16×10⁻¹² | 0.043 [0.035–0.051] | 1.01×10⁻²⁴ | 0.047 [0.039–0.054] | 3.34×10⁻³⁵ | 84 | 0.01 |
| | BMI (kg/m²) | 0.047 [0.032–0.062] | 2.12×10⁻⁰⁹ | 0.047 [0.041–0.054] | 8.89×10⁻⁴⁴ | 0.047 [0.041–0.053] | 4.27×10⁻⁵³ | 0 | 0.94 |
| | Born in South Asia (Yes/No) | 0.081 [−0.123 to 0.285] | 0.44 | 0.201 [0.133–0.269] | 8.15×10⁻⁰⁹ | 0.189 [0.124–0.253] | 1.01×10⁻⁰⁸ | 17 | 0.27 |
| | Parental history of T2D (Yes/No) | 0.122 [−0.015 to 0.258] | 0.08 | 0.138 [0.063–0.213] | 3.00×10⁻⁰⁴ | 0.134 [0.069–0.2] | 6.00×10⁻⁰⁵ | 0 | 0.83 |
| | Parity | −0.122 [−0.214 to −0.029] | 0.01 | −0.057 [−0.088 to −0.026] | 3.00×10⁻⁰⁴ | −0.063 [−0.093 to −0.034] | 2.00×10⁻⁰⁵ | 41 | 0.19 |
| | Education level (per level) | −0.045 [−0.106 to 0.016] | 0.15 | −0.045 [−0.075 to −0.014] | 0.004 | −0.045 [−0.072 to −0.017] | 0.001 | 0 | 0.99 |
| | Low diet quality (Yes/No) | 0.215 [0.064–0.366] | 0.005 | – | – | – | – | – | – |
| GDM (IADPSG criteria) | PRS (per 1 SD increase) | 1.56 [1.3–1.88] | 2.97×10⁻⁰⁶ | 1.42 [1.27–1.59] | 1.09×10⁻⁰⁹ | 1.45 [1.32–1.6] | 2.27×10⁻¹⁴ | 0 | 0.4 |
| | Age (year) | 1.13 [1.07–1.19] | 2.50×10⁻⁰⁶ | 1.1 [1.07–1.13] | 1.07×10⁻¹³ | 1.11 [1.08–1.13] | 1.98×10⁻¹⁸ | 0 | 0.4 |
| | BMI (kg/m²) | 1.08 [1.04–1.12] | 1.00×10⁻⁰⁴ | 1.08 [1.06–1.11] | 1.01×10⁻¹⁴ | 1.08 [1.06–1.1] | 6.25×10⁻¹⁸ | 0 | 0.87 |
| | Born in South Asia (Yes/No) | 1.35 [0.78–2.43] | 0.3 | 1.72 [1.35–2.19] | 1.00×10⁻⁰⁵ | 1.65 [1.33–2.06] | 8.37×10⁻⁰⁶ | 0 | 0.44 |
| | Parental history of T2D (Yes/No) | 1.67 [1.17–2.38] | 0.005 | 1.53 [1.21–1.94] | 5.00×10⁻⁰⁴ | 1.57 [1.29–1.92] | 7.06×10⁻⁰⁶ | 0 | 0.69 |
| | Parity | 0.86 [0.68–1.09] | 0.23 | 0.87 [0.79–0.95] | 0.003 | 0.87 [0.79–0.95] | 0.001 | 0 | 0.99 |
| | Education level (per level) | 0.89 [0.76–1.05] | 0.18 | 0.9 [0.81–0.99] | 0.04 | 0.9 [0.82–0.98] | 0.01 | 0 | 0.96 |
| | Low diet quality (Yes/No) | 1.68 [1.14–2.47] | 0.008 | – | – | – | – | – | – |

Models were additionally adjusted for the first five principal components (PCs) of each study. Abbreviations: BiB, Born in Bradford; BMI, Body mass index; CI, Confidence interval; GDM, Gestational diabetes mellitus; IADPSG, International Association of Diabetes and Pregnancy Study Groups; OR, Odds ratio; Q_E, P-value from the test for (residual) heterogeneity; SA, South Asia; SD, Standard deviation; START, South Asian birth cohort; T2D, Type 2 diabetes.

**Table 3.** Population attributable fractions of GDM risk factors in mothers from the START and Born in Bradford studies (multivariable models).

| Independent variable | START | | BiB | | Meta-analysis | | | |
|---|---|---|---|---|---|---|---|---|
| | AF [95% CI] | p Value | AF [95% CI] | p Value | AF [95% CI] | p Value | I² | $Q_E$ p value |
| Age (29–31 vs. <29 years) | 5.6 [−9.1 to 20.2] | 0.46 | 8.3 [3.5–13] | $6.00\times10^{-04}$ | 8 [3.5–12.5] | $5.00\times10^{-04}$ | 0 | 0.73 |
| Age (>32 vs. <29 years) | 31.2 [17.1–45.3] | $1.00\times10^{-05}$ | 20.2 [14.8–25.7] | $4.72\times10^{-13}$ | 21.7 [16.6–26.8] | $9.19\times10^{-17}$ | 50 | 0.16 |
| Body mass index (≥23 vs.<23) | 21.8 [8.7–34.9] | 0.001 | 33.8 [25.4–42.2] | $2.47\times10^{-15}$ | 30.3 [23.3–37.4] | $3.59\times10^{-17}$ | 56 | 0.13 |
| Born in SA (Yes vs. No) | 13.5 [−17.2 to 44.3] | 0.39 | 19.3 [12.6–26] | $1.47\times10^{-08}$ | 19 [12.5–25.6] | $1.07\times10^{-08}$ | 0 | 0.72 |
| Education (Post-secondary vs. less) | −18.2 [−46.8 to 10.5] | 0.21 | −0.8 [−4.6 to 3.1] | 0.7 | −1.1 [−4.9 to 2.7] | 0.58 | 28 | 0.24 |
| Parental history of T2D (Yes vs. No) | 15.1 [4.4–25.7] | 0.005 | 8.3 [4.1–12.5] | $1.00\times10^{-04}$ | 9.2 [5.3–13.1] | $3.54\times10^{-06}$ | 26 | 0.24 |
| PRS (Tertile 3 vs. 1+2) | 13.8 [4.9–22.6] | 0.002 | 12.2 [7.8–16.6] | $5.14\times10^{-08}$ | 12.5 [8.6–16.5] | $4.47\times10^{-10}$ | 0 | 0.76 |
| Low Diet Quality (Yes vs. No) | 8.9 [1.5–16.4] | 0.02 | – | – | – | – | – | – |
| Sum PAF of PRS (T3) and parental history of diabetes | 28.9 | | 20.5 | | 21.7 | | | |

GDM status derived using IADPSG criteria. Multivariate models included age, BMI, region of birth (South Asia vs other), education, parental history of diabetes, parity, principal components 1–5, and diet quality (START only) when applicable. Abbreviations: BiB, Born in Bradford; BMI, Body mass index, CI, Confidence interval; GDM, Gestational diabetes mellitus; IADPSG, International Association of Diabetes and Pregnancy Study Groups; PAF, Population attributable fraction; PRS, Polygenic risk score; $Q_E$ P, P-value from the test for (residual) heterogeneity; START, SouTh Asian BiRth CohorT.

detect a similar interaction to BiB of 9.9%). Subgroup analysis shows that the impact of a higher PRS on the risk of GDM was stronger in participants in lower BMI categories (*Supplementary file 1f*, *Figure 1*). Finally, a PRS×diet quality interaction on FPG was detected in START ($P_{interaction}$=0.002; *Table 4*), whereby the effect of the PRS appeared to be stronger in participants with a low diet quality (Beta=0.17 [95% CI=0.10–0.24]) than in participants with a medium or high diet quality (Beta=0.05 [95% CI=0.00–0.09]) (*Supplementary file 1f* and *Figure 2*). Our analysis shows that we have 90% power to detect such an interaction. The overall diet quality score was not available in BiB; hence, this interaction could not be tested for replication.

## Discussion

We demonstrate that a T2D PRS, based on an independent and multi-ethnic GWAS meta-analysis (with ~18% South Asian participants), is strongly associated with GDM and related glucose traits among South Asian pregnant women settled in Canada and the United Kingdom. This association is independent of other known GDM risk factors, including age, BMI, parental history of diabetes, and birth country. The PRS highest tertile accounted for 12.5% of the PAF of GDM. Consistent with a recent trans-ethnicity GWAS of GDM, and these results support the hypothesis that GDM and T2D are part of the same underlying pathology (*Pervjakova et al., 2022*).

Family history of T2D is often used as a surrogate marker of the genetic risk of T2D. Our results show that the addition of the PRS to the multivariate models does not nullify the impact of parental history on GDM and vice versa. This suggests that the PRS and family history of diabetes both partially convey independent information. This partial independence could be explained by the fact that the PRS does not entirely capture the genetic association signals with GDM. On the other hand, family history reflects not only genetic similarity, but also shared non-genetic lifestyle factors.

By deriving a T2D PRS and showing its significant association with the risk of GDM, we confirm that the two diseases share a substantial proportion of their genetic background. In their recent publication, *Pervjakova et al., 2022* also describe strong genetic similarities between the two traits by comparing the association and effect size of T2D variants to their effect on GDM. This convergence of observations using two different approaches (testing a PRS in our case versus independent loci in Pervjakova et al.) solidifies the hypothesis of a common genetic background between T2D and

**Table 4.** Interaction effects between GDM risk factors and T2D PRS in START and BiB.

| Interaction term | Dependent variable | START Beta/OR [95% CI]^a | START P_interaction | BiB Beta/OR [95% CI]^a | BiB P_interaction | Meta-analysis Beta/OR [95% CI]^a | Meta-analysis I² | Meta-analysis Q_E p value | Meta-analysis P_interaction |
|---|---|---|---|---|---|---|---|---|---|
| Fasting glucose | PRS×Age | −0.006 [−0.016 to 0.003] | 0.2 | 0.004 [0–0.008] | 0.07 | 0.002 [−0.001 to 0.006] | 72 | 0.23 | 0.06 |
| | PRS×BMI | **−0.01 [−0.019 to −0.002]** | **0.01** | 0.004 [0–0.008] | 0.05 | 0.001 [−0.002 to 0.005] | 89 | 0.42 | **0.002** |
| | PRS×Born in South Asia | **−0.137 [−0.268 to −0.006]** | **0.04** | 0.037 [−0.003 to 0.078] | 0.07 | 0.022 [−0.016 to 0.061] | 84 | 0.26 | **0.01** |
| | PRS×Parental history of T2D | −0.059 [−0.139 to 0.022] | 0.15 | 0.016 [−0.028 to 0.061] | 0.48 | −0.001 [−0.04 to 0.037] | 61 | 0.94 | 0.11 |
| | PRS×Parity | −0.014 [−0.062 to 0.034] | 0.56 | 0.004 [−0.01 to 0.018] | 0.6 | 0.002 [−0.011 to 0.016] | 0 | 0.73 | 0.48 |
| | PRS×Education level | −0.014 [−0.05 to 0.022] | 0.45 | −0.005 [−0.022 to 0.013] | 0.6 | −0.006 [−0.022 to 0.009] | 0 | 0.42 | 0.65 |
| | PRS×Low diet quality | **0.141 [0.053–0.228]** | **0.002** | – | – | – | – | – | – |
| 2 hr post-load glucose | PRS×Age | 0 [−0.03 to 0.03] | 0.98 | 0.01 [−0.001 to 0.021] | 0.07 | 0.009 [−0.001 to 0.019] | 0 | 0.08 | 0.54 |
| | PRS×BMI | −0.022 [−0.047 to 0.003] | 0.09 | 0 [−0.01 to 0.011] | 0.94 | −0.003 [−0.012 to 0.007] | 61 | 0.56 | 0.11 |
| | PRS×Born in South Asia | −0.191 [−0.586 to 0.205] | 0.34 | 0.072 [−0.039 to 0.182] | 0.2 | 0.053 [−0.054 to 0.159] | 36 | 0.33 | 0.21 |
| | PRS×Parental history of T2D | −0.092 [−0.335 to 0.151] | 0.46 | 0.055 [−0.066 to 0.177] | 0.37 | 0.026 [−0.083 to 0.135] | 11 | 0.64 | 0.29 |
| | PRS×Parity | −0.039 [−0.184 to 0.107] | 0.6 | 0.009 [−0.03 to 0.047] | 0.66 | 0.005 [−0.032 to 0.043] | 0 | 0.77 | 0.54 |
| | PRS×Education level | 0.037 [−0.072 to 0.146] | 0.51 | 0.008 [−0.039 to 0.056] | 0.73 | 0.013 [−0.031 to 0.056] | 0 | 0.56 | 0.64 |
| | PRS×Low diet quality | 0.068 [−0.199 to 0.335] | 0.62 | – | – | – | – | – | – |
| AUC glucose | PRS×Age | −0.007 [−0.023 to 0.009] | 0.41 | 0.004 [−0.002 to 0.011] | 0.19 | 0.003 [−0.003 to 0.009] | 37 | 0.36 | 0.21 |
| | PRS×BMI | **−0.014 [−0.027 to 0]** | **0.05** | 0.002 [−0.004 to 0.008] | 0.52 | −0.001 [−0.006 to 0.005] | 77 | 0.82 | **0.04** |
| | PRS×Born in South Asia | −0.126 [−0.34 to 0.088] | 0.25 | 0.015 [−0.051 to 0.081] | 0.65 | 0.003 [−0.06 to 0.066] | 35 | 0.93 | 0.22 |
| | PRS×Parental history of T2D | −0.027 [−0.158 to 0.105] | 0.69 | 0.025 [−0.048 to 0.098] | 0.49 | 0.013 [−0.051 to 0.077] | 0 | 0.68 | 0.5 |
| | PRS×Parity | −0.057 [−0.135 to 0.022] | 0.16 | 0.006 [−0.017 to 0.029] | 0.6 | 0.001 [−0.021 to 0.023] | 56 | 0.91 | 0.13 |
| | PRS×Education level | 0.007 [−0.052 to 0.066] | 0.82 | −0.008 [−0.036 to 0.021] | 0.6 | −0.005 [−0.03 to 0.021] | 0 | 0.71 | 0.67 |
| | PRS×Low diet quality | 0.07 [−0.074 to 0.214] | 0.34 | – | – | – | – | – | – |
| GDM (IADPSG criteria) | PRS×Age | 0.99 [0.94–1.03] | 0.59 | 0.99 [0.96–1.01] | 0.17 | 0.99 [0.97–1] | 0 | 0.14 | 0.92 |
| | PRS×BMI | 0.97 [0.94–1.01] | 0.15 | **0.98 [0.96–1]** | **0.03** | **0.98 [0.96–0.99]** | 0 | 0.01 | 0.76 |
| | PRS×Born in South Asia | 0.65 [0.33–1.23] | 0.2 | 1.04 [0.82–1.31] | 0.76 | 0.98 [0.79–1.23] | 41 | 0.89 | 0.19 |
| | PRS×Parental history of T2D | 0.72 [0.5–1.04] | 0.08 | 1.07 [0.85–1.35] | 0.59 | 0.95 [0.78–1.16] | 67 | 0.63 | 0.08 |
| | PRS×Parity | 0.88 [0.71–1.09] | 0.23 | 0.99 [0.92–1.07] | 0.86 | 0.98 [0.91–1.05] | 15 | 0.57 | 0.28 |
| | PRS×Education level | 1.01 [0.85–1.19] | 0.93 | 0.93 [0.85–1.02] | 0.12 | 0.95 [0.87–1.03] | 0 | 0.2 | 0.4 |
| | PRS×Low diet quality | 1.26 [0.85–1.89] | 0.26 | – | – | – | – | – | – |

Nominally significant results are shown in bold. Results from models adjusted for age, BMI, education level, birth region (South Asia vs. other), parity, parental history of diabetes, and genetic PC axes 1–5. ^a Values are Beta for continuous dependent variables (fasting 2 hr, and AUC glucose), and OR for binary dependent variable (i.e. GDM). Abbreviations: AUC, area under the curve; BiB, Born in Bradford; BMI, body mass index; CI, confidence interval; GDM, gestational diabetes mellitus; IADPSG, International Association of Diabetes and Pregnancy Study Groups; OR, odds ratio; PRS, polygenic risk score; Q_E, P-value from the test for (residual) heterogeneity; START, SouTh Asian BiRth CohorT.

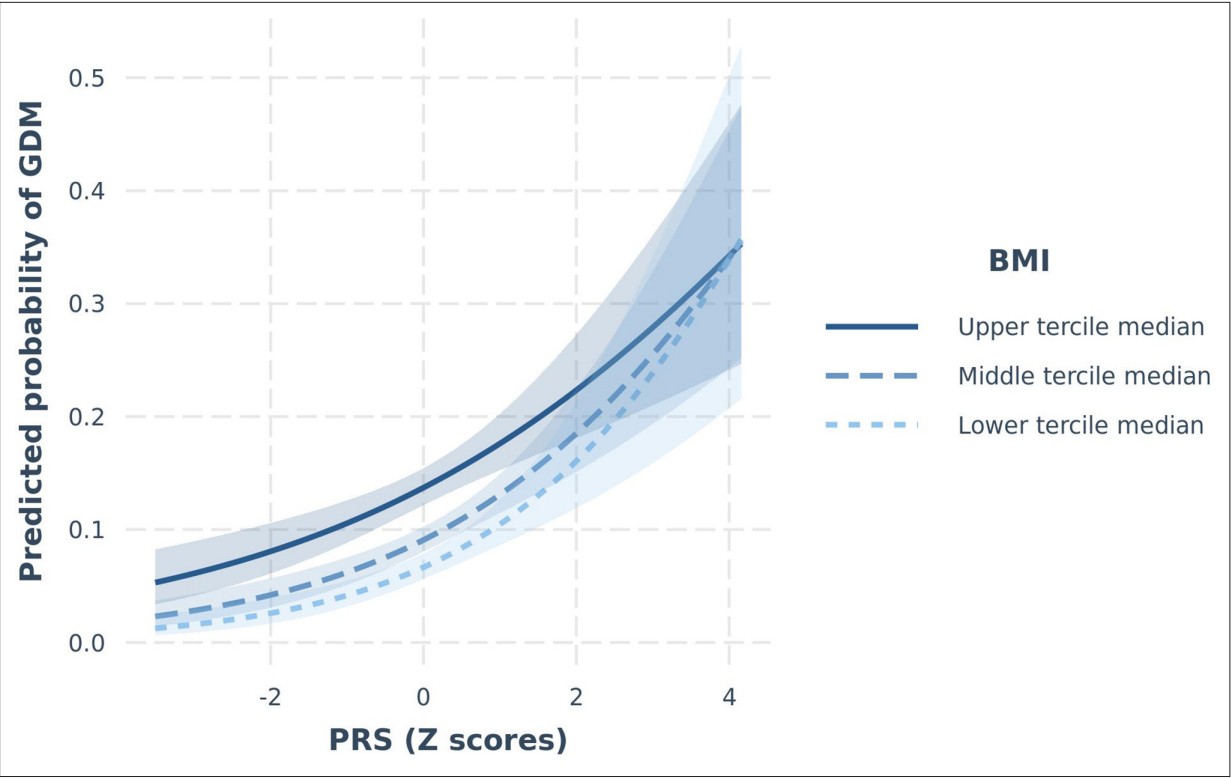

**Figure 1.** Predicted probability of GDM_{IADPSG} as a function of PRS (continuous), stratified by BMI groups in BiB. Lines (with 95% confidence limits) represent predicted probabilities of GDM stratified by BMI groups (upper, middle, and lower terciles). Models are adjusted for maternal age. BiB, Born in Bradford; BMI, body mass index; GDM, gestational diabetes mellitus; IADPSG, International Association of Diabetes and Pregnancy Study Groups; PRS, polygenic risk score; SD, standard deviation.

GDM. It is however important to note that, although BiB's South Asian mothers were included in both analysis, they represented ~1.2% of the total sample size in Pervjakova et al., which suggests that our congruent conclusions are unlikely to have been driven by the sample overlap between the two studies.

Overall, the evidence for modulation of the PRS's effect on GDM-related traits by other GDM risk factors was weak. Most interactions tested were not significant in both studies. This absence of significance should however be treated with caution since our power analysis suggests that, given our sample size, we are only able to detect strong interaction effects. Two marginal PRS×BMI and PRS×-South Asia born interactions on FPG were observed, these were close to significance in both studies but did not replicate definitively, both in terms effect sizes and direction of effect, which precludes a power issue, and suggests differences in the effect of these environmental factors between the two studies, or possibly false positive results. Furthermore, these interactions would not pass multiple testing corrections if applied. Two potentially stronger PRS×diet quality, and PRS×BMI interactions modulating FPG and GDM were observed in START, and BiB respectively. However, since it was not possible to replicate these interactions (i.e., no comparable diet data available in BiB, and low power in START), future investigations are required in order to validate these observations. If confirmed, these interactions may help identify a subpopulation who will benefit the most from a targeted intervention for the prevention of GDM. Given the transient nature of GDM, another important research question would be the identification of women at greater risk of developing T2D after developing GDM, and how the genetic risk modulates this progression. This could be done by testing the interactions between a GDM/T2D PRS and T2D status in women with prior GDM. This could reveal whether women with prior GDM and a high genetic risk are more likely to develop T2D than women with prior GDM and a low genetic risk. Finally, given the low sensitivity of the PRS themselves, future studies should focus on deriving and estimating the predictive value of a composite score which combines the

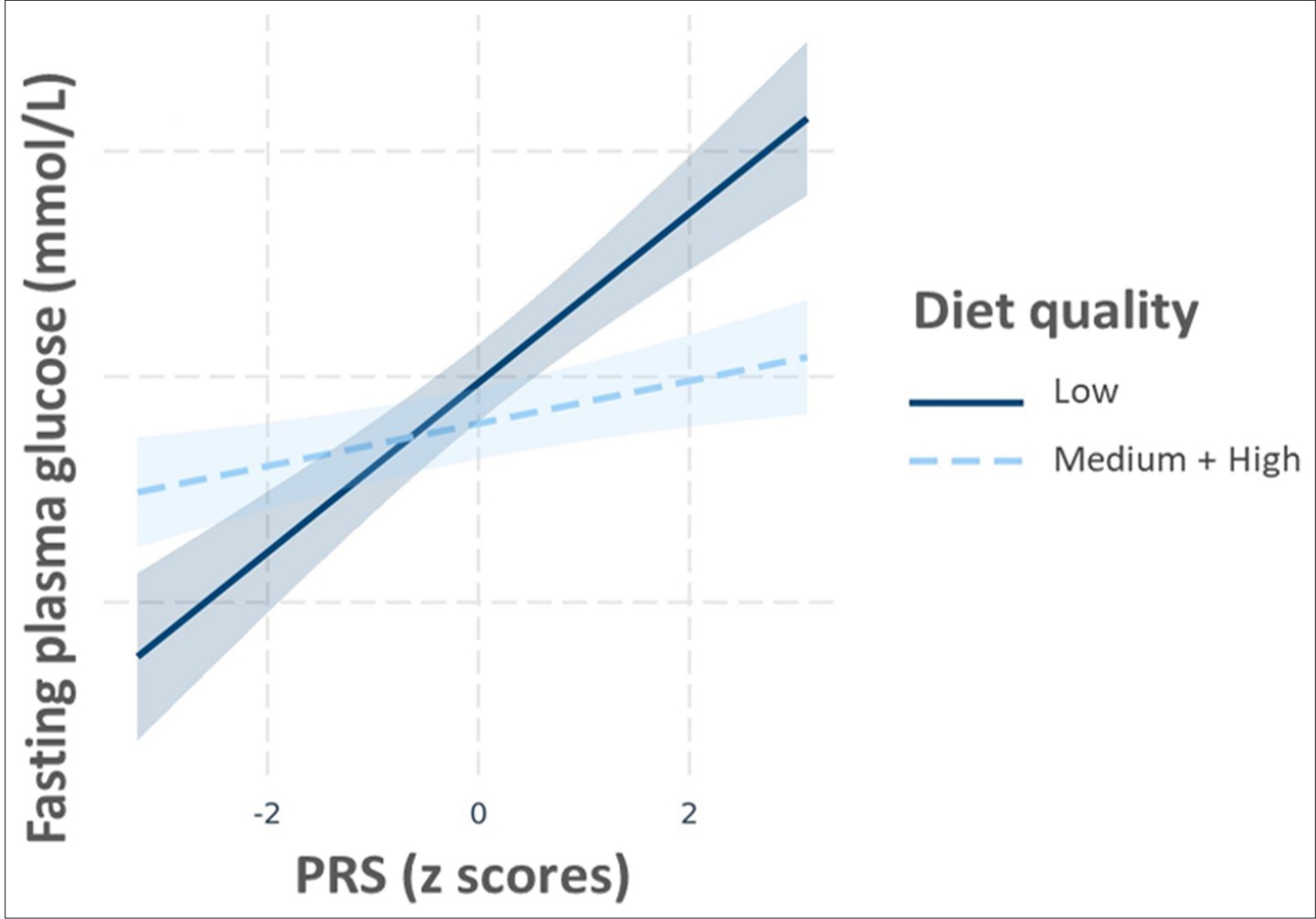

**Figure 2.** Multivariable regression of PRS (continuous) fasting plasma glucose (FPG) stratified by diet quality score in START. Regression lines (with 95% confidence limits) represent predictions of FPG. Models are adjusted for maternal age, and BMI. BMI, body mass index; PRS, polygenic risk score; START, SouTh Asian BiRth CohorT.

GDM/T2D PRS, family history of diabetes, prior GDM status, and diet quality score in order to improve the identification of women at higher risk of developing T2D.

The overall clinical implications of our findings should be carefully considered. At present, the use of laboratory-derived genetic information in the clinical setting remains expensive and is not implemented for complex diseases like GDM or T2D. Furthermore, our results show that, despite a strong association, the PRS has a low discriminatory value (detection rate of 10% for a 5% false positive rate) regarding GDM cases. This is in line with the observations of *Wald and Old, 2019* stating that most polygenic scores of complex traits derived to date would perform poorly as a screening tests in a clinical setting.

Our study has been considerably strengthened by the use of a PRS optimized for a large population of South Asians from two independent cohorts, as well as by the fact that GDM status was determined using objective OGTT measures. Nevertheless, there are some limitations to our analysis that should be considered: (i) the weights attributed to the genetic variants included in the PRS are derived from a T2D study. Overall, evidence points to a strong correlation between top variants from T2D and GDM GWASs. However, variants at some common loci (e.g., *MTNR1B*) might have significantly different effect size depending on the phenotype studied (*Pervjakova et al., 2022*). In addition, variants in at least one locus (*HKDC1*) have been strongly associated to GDM but not T2D (*Pervjakova et al., 2022*). More GDM-specific loci, or loci with a different magnitude of effect between GDM and T2D might be identified from future, larger studies. These observations suggest that future PRSs

based on a GDM GWAS may have more power to detect gene×environment interactions. (ii) Second, some differences in measurements exist between START and BiB studies, including the timing of weight measurements, and the number of data points included in the calculation of AUCg. However, since data were standardized in both studies, we do not expect that AUCg measurements differences had a major impact on the results. (iii) Finally, the comparison of genetic data between START and BiB revealed the existence of slight genetic heterogeneity, both between and within the samples of these two cohorts. It is our assumption that these differences can be explained by the difference of sample size (START being smaller than BiB), as well as by historical differences in migration patterns from South Asia to Canada and the United Kingdom. For example, most START participants were first-generation migrants from India, whereas the majority of South Asians in BiB are descendants of Pakistani migrants who settled in the United Kingdom for several generations. In order to account for this genetic heterogeneity, we derived our T2D PRS by combining samples from the two studies. This PRS should be more generalizable to other South Asian studies. Another measure implemented to reduce the effect of population stratification was the adjustment for the PC axes in our analysis. Given the absence of heterogeneity in our FPG, 2hG, or GDM$_{IADPSG}$ PC adjusted models, we consider that population stratification effects have been accounted for.

## Conclusion

A T2D-derived PRS is strongly associated with the risk of GDM in pregnant women of South Asian descent, independent of parental history of diabetes, and other GDM risk factors.

## Methods

### Study design and participants

START is a prospective cohort study designed to evaluate the environmental and genetic determinants of cardio-metabolic traits among South Asian women and their offspring living in Canada (*Anand et al., 2013*). In brief, 1012 South Asian pregnant women, aged between 18 and 40 years old, were recruited during their second trimester of pregnancy from the Peel Region (Ontario, Canada) through physician referrals between 2011 and 2015. All START participants provided informed consent, and the study was approved by local ethics committees (Hamilton Integrated Research Ethics Board [ID:10-640], William Osler Health System [ID:11-0001], and Trillium Health Partners [RCC:11-018, ID:492]).

BiB is a prospective, longitudinal family cohort study designed to investigate the causes of illness, and develop interventions to improve health in a deprived multi-ethnic population in Bradford, England, UK (*Wright et al., 2013*). Between 2007 and 2011, 12,453 women of various ethnic back-grounds (~46% South Asian origin) were recruited between their 24th and 28th week of pregnancy. Detailed information on socio-economic characteristics, ethnicity, family history, environmental, and physical risk factors has been collected (*Farrar et al., 2015*; *Wright et al., 2013*). Ethical approval for all aspects of the research was granted by Bradford Research Ethics Committee [Ref 07/H1302/112].

### Measurements and questionnaires

#### SouTh Asian BiRth CohorT

A detailed description of the maternal measurements has been published previously (*Anand et al., 2017*). Briefly, weight and height were measured using standard procedures, and information about pre-pregnancy weight, family, and personal medical history was collected using questionnaires. Parental history of diabetes was derived from baseline questionnaires and categorized as neither parent, or either one, or both parents had a history of diabetes. Birth country, number of years spent in Canada, and education-related variables were self-reported. Participants' highest level of education was coded as a five-category ordinal variable as: 1—less than high school; 2—high school completed; 3—Diploma or certificate from trade, technical or vocational school; 4— Bach-elor's or undergraduate degree, or teacher's college; and 5— Master's, Doctorate or professional degree. A binary 'born in South Asia' variable was categorized as participants born in South Asia (India, Pakistan, Sri Lanka, or Bangladesh versus participants were born in any other country). A validated ethnic-specific food frequency questionnaire (FFQ) was used to collect dietary informa-tion (*Kelemen et al., 2003*). The following steps were implemented in order to calculate the diet quality of each participant: (i) for each of the following four food groups (green leafy vegetables;

raw vegetables; other cooked vegetables; and fruits), 1 point was given for consuming ≥the study population median (vs. 0 points if intake <population median); (ii) for each of the following two food groups (fried foods/fast food/snacks; and meat/poultry), 1 point was given for consuming <the study population median (vs. 0 points if intake ≥population median); (iii) the points attributed to each of the six food groups mentioned above were summed in order to derive a continuous food score (ranging from 0 to 6 points), which was subsequently divided into three categories (Low diet quality — if food score=1 or 2; Medium diet quality — if food score=3 or 4; and High diet quality if food score=5 or 6). (iv) A binary diet quality variable used in our analysis was coded as follows (Low diet quality — if food score=1 or 2; medium or high quality — if food score≥3) (*Anand et al., 2017*).

### Born in Bradford

Maternal height was measured during the recruitment visit (24–28th weeks of pregnancy) using standard procedures. In the absence of pre-pregnancy weight data, weight from the first antenatal clinic visit (average 12 weeks of pregnancy) was used to calculate BMI. Ethnicity of participants and years spent in the United Kingdom were self-reported at recruitment through an interview administered questionnaire; missing ethnicity data were backfilled from primary care data when available. The South Asian ethnicity of all participants included in this analysis was validated using genetic data. Parental history of diabetes and 'born in South Asia' variables were derived from the baseline questionnaire data and coded as in START. Since only a very small proportion of BiB's participants completed an FFQ that included information about fruits and vegetables intake, the diet quality score could not be derived in BiB. Data regarding the participant's highest educational qualification were equalized (using UK standards) and recoded into the following categories: 1— less than 5 General Certificate of Secondary Education (GCSE) equivalent; 2— 5 GCSE equivalent; 3— A-level equivalent; and 4— higher than A-level. Data for unclassifiable foreign degrees were considered as missing.

## Outcomes

Study participants without prior T2D were invited to undertake a 75-g oral glucose tolerance test (OGTT) in both START and BiB, and FPG, and 2hG levels were measured (1 hr post-load glucose was measured in START only). AUCg was calculated using the FPG and 2hG glucose levels in BiB, and using the FPG, 1 hr post-load glucose, and 2hG levels in START (*Anand et al., 2017*). Given the difference in the number of data points included in the calculation of AUC between the two studies and the skewness of the distributions, values were log-transformed, winsorized, and standardized in each study before analysis. Gestational diabetes status of women without pre-existing T2D was primarily defined based on OGTT results in both studies using the International Association of Diabetes and Pregnancy Study Group (IADPSG) GDM criteria (FPG≥5.1 mmol/L or higher, or a 1hG≥10.8 or a 2hG≥8.5 mmol/L or higher) (*Metzger et al., 2010*). Our secondary outcome was GDM using BiB's South Asian specific definition (FPG of 5.2 mmol/L or higher, or a 2hG of 7.2 mmol/L or higher) (*Farrar et al., 2015*), which will be referred to as the South Asian-specific definition hereafter. Self-reported GDM status or data from the birth chart were used to determine GDM's status if OGTT measures were unavailable (N=65 and 31 in START and BiB, respectively). Women with pre-existing diabetes at baseline were not included in this analysis. Pre-pregnancy diabetes status was determined using maternal self-reported data (about diabetes diagnosis, diabetes medication, and/or insulin intake prior to pregnancy) in START. In BiB, information on pre-pregnancy diabetes was backfilled from electronic medical records.

In order to keep a single pregnancy (and a single GDM status) per mother in BiB, only pregnancies with no missing data for GDM were included. For mothers with available data at multiple pregnancies at this stage, pregnancies with no missing data across all covariates (age, BMI, family history, birth country, parity, and education level) were prioritized. Next, only pregnancies with the least amount of missing data across all covariates were kept. The following two additional filtering approaches were then applied for mothers with multiple pregnancies remaining: (i) if GDM was not diagnosed at any of the pregnancies, phenotype data at the latest available time point was kept (i.e., keep older GDM controls) and (ii) if GDM was diagnosed during any of the pregnancies included in the study, the earliest time point where GDM was diagnosed was kept (i.e., keep younger GDM cases).

## DNA extraction, genotyping, imputation, and filtering

### SouTh Asian BiRth CohorT

DNA was extracted and genotyped for 867 mothers using the Illumina Human CoreExome-24 and Infinium CoreExome-24 arrays (Illumina, San Diego, CA). About 837 samples passed standard quality control procedures (*Anderson et al., 2010*). Genotype data was handeled using PLINK v1.90b6.8 (*Chang et al., 2015*) . Genotypes were phased and imputed using SHAPEIT v2.12 (*Delaneau et al., 2014*), and IMPUTE v2.3.2 (*Howie et al., 2009*), respectively, using the 1000 Genomes (phase 3) data as a reference panel (*Auton et al., 2015*). Variants with an info score <0.7 were removed from analysis. In total, 837 START participants with both genotypes and available GDM status, FPG, 1hG, and/or 2hG levels were included in the analysis.

### Born in Bradford

DNA was extracted and genotyped for 16,267 and 3663 BiB participants using the Illumina Human-CoreExome (12v1.0, 12v1.1, or 24v1.0) and InfiniumGlobal Screening Array (24v2.0) arrays, respectively (Illumina, San Diego, CA). About 4372 South Asian mothers passed genotyping quality controls, had GDM status, FPG, and/or 2hG levels available, and were included in our analysis. Genotype data was handeled using PLINK v1.90b6.8 (*Chang et al., 2015*).

## Deriving the PRS

Given the absence of publicly available South Asian-specific T2D or GDM GWAS data at the time of the analysis, weights were derived from the DIAGRAM's 2014 multi-ethnic T2D GWAS meta-analysis, which included over 18% of South Asians (~63% European and 19% other ethnic backgrounds) (*Mahajan et al., 2014*). A grid search approach was used to identify the optimal parameters (17 p values tested, ranging from $5 \times 10^{-8}$ to 1 with 0.1 increase; 4 heritability values tested: 0.023, 0.06, 0.08, and 0.12). START and BiB genotypes were pooled. About 70% of the samples' data were used for training and 30% for validation (random sampling stratified by study) in order to minimize the impact of population stratification. The PRS was derived using LDpred2 (*Privé et al., 2020*). The best PRS (i.e., that maximized the AUC) was characterized by a p value≤0.0014 and an $h^2$=0.08 ($N_{SNVs}$=6492). The PRS was standardized (mean=0, standard deviation=1) in both studies before analysis.

## Principal component analysis of genetic data

A principal component analysis (PCA) was performed using the PC-Air function from the GENESIS R package (v2.20.0) (*Conomos et al., 2015a*; *Conomos et al., 2015b*). Kinship matrices (required to derive PCs with PC-Air) were derived using KING (v2.2.5) (*Manichaikul et al., 2010a*; *Manichaikul et al., 2010b*).

## Statistical analysis

### Regression models

The statistical analysis was conducted using R (v3.6.3) (R core Team, 2016). Linear regression models were used to test the association between the PRS and FPG, 2hG and AUCg. PRS and GDM associations were tested using logistic regression. Both univariate and multivariate models were constructed with adjustment for GDM risk factors (age, BMI, parity, birth in South Asia [yes vs. no]), education level, and diet quality (in START only) and the first five PCs (in order to minimize the effect of population stratification). Interactions between the PRS and each risk factor was also tested. Interaction plots were produced using the interactions R package (v1.2.0.9000) (*Long, 2021*).

### Population attributable fractions

The estimated PAFs and their corresponding standard errors were calculated using the AF R package (v.0.1.5) (*Dahlqwist and Sjolander, 2019*). To this end, continuous variables were recoded into categorical variables: age was divided into two categories ([29–31, 32–43] vs. 19–28); BMI was stratified into a two categories variable using South Asian obesity cutoff points suggested by *Gray et al., 2011* (<23 vs. ≥23); the PRS was divided into two categories (tertiles 1+2 versus tertile 3); parity was divided into two categories (primiparity versus 1 pregnancy or more); education level was divided into two

categories (completed high school or lower versus higher degree, diploma, or certificate in START; and A-level equivalent or lower versus higher than A-level in BiB).

## Detection and false positive rates

Detection rate (sensitivity) and false positive rate (1-specificity) for the OR of association of PRS tertile 3 versus 1 was estimated using the risk-screening converter tool developed by *Wald and Morris, 2011*.

## Power analysis for interactions

Power to detect interactions was estimated using the InteractionPoweR R package (v0.1.1) (*Baranger et al., 2022*). Monte-carlo simulation was used using 10.000 simulations and an alpha of 0.05.

## Acknowledgements

Research projects in START and Born in Bradford are only possible because of the enthusiasm and commitment of the parents and children involved in these two studies. The authors are grateful to all the participants, teachers, school staff, health professionals, and researchers, and other contributors who have made these studies happen. Studies: The South Asian Birth Cohort (START) study data were collected as part of a program funded by the Indian Council of Medical Research in Canada and by the Canadian Institutes of Health Research (Grant INC-109205), and the Heart and Stroke Foundation (Grant NA7283) with founding principal investigators: Sonia S Anand, Anil Vasudevan, Milan Gupta, Katherine Morrison, Anura Kurpad, Koon K Teo, and Krishnamachari Srinivasan. The Born in Bradford (BiB) The Born in Bradford cohort is funded by the National Institute for Health Research Collaboration for Applied Health Research and Care (NIHR CLAHRC) and the Programme Grants for Applied Research funding scheme (RP-PG-0407-10044). The study also receives funding from the Wellcome Trust (WT101597MA), a joint grant from the UK Medical Research Council (MRC) and Economic and Social Science Research Council (ESRC) (MR/N024397/1) and the British Heart Foundation (CS/16/4/32482). DNA extraction was funded by the UK Medical Research Council via the Integrative Epidemiology Unit (MRC IEU; MC_UU_12013/5) and genotyping via the MRC IEU and a National Institute of Health Research Senior Investigator Award to DAL (NF-0616-10102). Research associate (AL) and graduate student (JL) costs were covered by two Canadian Institutes of Health Research Grants [Project grant number: 298104, Foundation Scheme grant number: FDN-143255, Study grant numbers: INC 109205, NA 7283] awarded to SSA; DAL's contribution to this study is supported by the Bristol NIHR Biomedical Research Centre, the UK Medical Research Council (MC_UU_00011/6) and the British Heart Foundation (CH/F/20/90003). SSA is supported by a Tier 1 Canada Research Chain in Ethnic Diversity and Cardiovascular Disease, and a Heart and Stroke Foundation/Michael G DeGroote Chair in Population Health Research at McMaster University.

## Additional information

### Competing interests

Deborah A Lawlor: has received support from Medtronic Ltd and Roche Diagnostics for research unrelated to that presented here. No financial relationships with any organisations that might have an interest in the submitted work in the previous three years; no other relationships or activities that could appear to have influenced the submitted work. The other authors declare that no competing interests exist.

### Funding

| Funder | Grant reference number | Author |
| --- | --- | --- |
| Canadian Institutes of Health Research | 298104 | Sonia S Anand |
| Canadian Institutes of Health Research | FDN-143255 | Sonia S Anand |

| Funder | Grant reference number | Author |
|---|---|---|
| Bristol NIHR Biomedical Research Center | | Deborah A Lawlor |
| UK Medical Research Council | MC_UU_00011/6 | Deborah A Lawlor |
| British Heart Foundation | CH/F/20/90003 | Deborah A Lawlor |
| Canada Research Chairs | | Sonia S Anand |
| Heart and Stroke Foundation | Michael G. DeGroote Chair | Sonia S Anand |
| Wellcome | WT101597MA | Deborah A Lawlor John Wright |
| Medical Research Council | MR/N024397/1 | Deborah A Lawlor John Wright |
| Economic and Social Research Council | MR/N024397/1 | Deborah A Lawlor John Wright |

The funders had no role in study design, data collection and interpretation, or the decision to submit the work for publication. For the purpose of Open Access, the authors have applied a CC BY public copyright license to any Author Accepted Manuscript version arising from this submission.

## Author contributions

Amel Lamri, Data curation, Software, Formal analysis, Validation, Investigation, Visualization, Methodology, Writing - original draft, Writing – review and editing; Jayneel Limbachia, Formal analysis, Writing - original draft, Writing – review and editing; Karleen M Schulze, Brian Kelly, Data curation, Writing – review and editing; Dipika Desai, Conceptualization, Project administration, Writing – review and editing; Russell J de Souza, Conceptualization, Writing – review and editing; Guillaume Paré, Supervision, Validation, Writing – review and editing; Deborah A Lawlor, John Wright, Resources, Funding acquisition, Writing – review and editing; Sonia S Anand, Conceptualization, Resources, Supervision, Funding acquisition, Writing - original draft, Project administration, Writing – review and editing

## Author ORCIDs

Amel Lamri http://orcid.org/0000-0001-7182-0661
Guillaume Paré http://orcid.org/0000-0002-6795-4760
Sonia S Anand http://orcid.org/0000-0003-3692-7441

## Ethics

Human subjects: All START and BiB participants provided informed consent. The START study was approved by local ethics committees (Hamilton Integrated Research Ethics Board [ID:10-640], William Osler Health System [ID:11-0001], and Trillium Health Partners [RCC:11-018, ID:492]). Ethical approval for all aspects of the research was granted by Bradford Research Ethics Committee [Ref 07/H1302/112].

## Decision letter and Author response

Decision letter https://doi.org/10.7554/eLife.81498.sa1
Author response https://doi.org/10.7554/eLife.81498.sa2

# Additional files

## Supplementary files

• Supplementary file 1. Supplementary tables. (a) Information on the variants included in the polygenic risk score. Genomic positions corresponds to GRCh37/hg19 assembly. SNV IDs, alleles, ORs and P-values are from *Mahajan et al., 2014* (Main text reference 30). Abbreviations: CI, confidence interval; OR, odds ratio; PRS, polygenic risk score. (b) Association results between PRS and GDM-related variables in START and BiB (Univariate models). [a] Values are Betas for continuous variables (fasting 2 h, and AUC glucose), and ORs for binary variable (i.e. GDM). Abbreviations: BiB, Born in Bradford; CI, confidence interval; IADPSG, International association of the diabetes and pregnancy study groups; OR, odds ratio; PRS, polygenic risk score; $Q_E$ P, P-value from the test

for (residual) heterogeneity; SE, standard error; START, South Asian birth cohort; T1, tertile 1; T2, tertile 2; T3, tertile3. (c) Association between GDM risk factors and GDM (South Asian-specific definition): results from multivariate models in START and BiB cohorts. Models were additionally adjusted for the first 5 PCs of each study. [a] Values are Betas for continuous variables (fasting 2 h, and AUC glucose), and ORs for binary variable (i.e. GDM). Abbreviations: BiB, Born in Bradford; BMI, Body mass index; CI, Confidence interval; GDM, Gestational diabetes mellitus; OR, Odds ratio; SA, South Asia; SD, Standard deviation; $Q_E$ P, P-value from the test for (residual) heterogeneity; START, South Asian birth cohort; T2D, Type 2 diabetes. (d) Association between categorical PRS (tertiles) and GDM-related traits in multivariate models. The table shows results for tertile 2 vs. tertile 1, and tertile 3 vs. tertile 1 [a] Values are Betas for continuous variables (fasting 2 h, and AUC glucose), and ORs for binary variables (i.e. GDM). Abbreviations: BiB, Born in Bradford; CI, confidence interval; IADPSG, International association of the diabetes and pregnancy study groups; OR, odds ratio; PRS, polygenic risk score; $Q_E$ P, P-value from the test for (residual) heterogeneity; SE, standard error; START, South Asian birth cohort; T1, tertile 1; T2, tertile 2; T3, tertile3. (e) Interaction effects between GDM risk factors and T2D PRS on GDM (South Asian-specific definition) in START and BiB. Results from models adjusted for age, BMI, education level, birth region (South Asia vs. other), parity, parental history of diabetes, and genetic PC axes 1–5. [a] Values are Beta for continuous dependent variables (fasting 2 h, and AUC glucose), and OR for binary variable (i.e. GDM). Abbreviations: AUC, area under the curve; BiB, Born in Bradford; BMI, body mass index; CI, confidence interval; GDM, gestational diabetes mellitus; OR, odds ratio; PRS, polygenic risk score; $Q_E$ P, P-value from the test for (residual) heterogeneity; START, South Asian birth cohort. (f) Association of PRS with fasting glucose levels and GDM by BMI, birth country (South Asia vs. others), and diet quality categories in START and BiB. Results from models adjusted for age, education level, parity, and genetic PC axes 1–5, birth region (South Asia vs. other) (rows 1–3), and BMI (rows 4–7). [a] Values are Beta for continuous dependent variables (i.e. fasting glucose), and OR for binary variables (i.e. GDM). Abbreviations: BiB, Born in Bradford; BMI, body mass index; CI, confidence interval; GDM, gestational diabetes mellitus; IADPSG, International association of the diabetes and pregnancy study groups; PRS, polygenic risk score; SE, Standard error; START, South Asian birth cohort.

- MDAR checklist

## Data availability

Data from START is not publicly available, since the study is bound by consent which indicates the data will not be used by an outside group. Requests for collaboration or replication will be considered for research purposes only (no commercial use allowed, as per the study's informed consent). Requests should be addressed to the study's principal investigator (Sonia Anand, anands@mcmaster. ca) via a form which will be provided upon request by emailing natcampb@mcmaster.ca. The request will be evaluated by PIs and co-investigators, and projects deemed of scientific interest will be further evaluated/validated by local REB chair. Born in Bradford data are available for research purposes only by sending an expression of interest form downloadable from https://borninbradford.nhs.uk/wp-content/uploads/BiB_EoI_v3.1_10.05.21.doct to borninbradford@bthft.nhs.uk . The proposal will be reviewed by BiB's executive team. If the request is approved, the requester will be asked to sign a Data Sharing Contract and a Data Sharing Agreement. Full details on how to access data and forms can be found here https://borninbradford.nhs.uk/research/how-to-access-data/. The code used to analyze the data is available at https://github.com/AmelLamri/Paper_T2dPrsGdm_StartBiB (copy archived at swh:1:rev:78a26e8d3c4088325572b8a79e132dca65b7a67f). All Sharable processed versions of the datasets used in the manuscript are made available as supplementary material or at https://github.com/AmelLamri/Paper_T2dPrsGdm_StartBiB.

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
