## [Editor Report]

South Asian women have twice the risk of developing Gestational Diabetes Mellitus (GDM) compared with white European women. This clearly presented comprehensive study shows that a T2D polygenic risk score is strongly associated with multiple GDM-related traits in South Asian women and is a significant contributor to the population-attributable fraction of GDM, independently of family history of diabetes. This will be of interest to genetic epidemiologists and clinicians working in this field.

---

## [Decision Letter]

**Decision letter after peer review:**

Thank you for submitting your article "The Genetic Risk of Gestational Diabetes in South Asian women" for consideration by *eLife*. Your article has been reviewed by 2 peer reviewers, including Edward D Janus as Reviewing Editor and Reviewer #1, and the evaluation has been overseen by Ricardo Azziz as the Senior Editor.

Essential revisions:

1) Given that after delivery GDM resolves to normoglycaemia in most cases there is considerable interest in finding ways to determine which women with GDM (rather than all) should be targeted for diet and lifestyle intervention to prevent development of subsequent T2DM. While this is not the objective of your study you could nevertheless note this in your discussion and suggest how this might be studied to further understanding of this issue.

2) Are the 'Born in Bradford' participants in this study also in the Pervjakova paper? – Pervjakova et al. 2022, PMID: 35220425 which is referenced in the manuscript (as a medRxiv – now published). Please make clear what the degree of overlap between BIB participants is between these papers if any. Please be more explicit about what this paper adds to the Pervjakova analysis in the Discussion. This will be useful to readers, because that previous paper reached similar conclusions about the overlap between T2D and GDM genetic susceptibilities, and most of the statistical power here comes from the BIB cohort.

3) The support for the hypothesis from the newly reported START cohort is strong and important. It seems unlikely that the interaction analyses would have had adequate power for the negative result to be interpreted as a strong negative. Can the authors comment on this, and possibly quantify the power they had?

4) There is a debate about the utility of PRS in screening which goes beyond cost (p16); as the authors will know Wald and others have argued that for most diseases PRS, even with a high aetiological fraction, would lack sensitivity and specificity if used for screening. Readers might be interested to know what sort of sensitivity and specificity for GDM might result with the OR of 1.47 found here. The authors do use the phrase "enhanced predictive value" on p16 so I believe asking for that to be quantified is fair.

---

## [Author Response]

Essential revisions:1) Given that after delivery GDM resolves to normoglycaemia in most cases there is considerable interest in finding ways to determine which women with GDM (rather than all) should be targeted for diet and lifestyle intervention to prevent development of subsequent T2DM. While this is not the objective of your study you could nevertheless note this in your discussion and suggest how this might be studied to further understanding of this issue.

Thank you for this suggestion. This point is now addressed in the discussion page 11 line 167 of the revised.docx manuscript (line 168 in the.pdf file) where we state that:

“Given the transient nature of GDM, another important research question would be the identification of women at greater risk of developing T2D after developing GDM, and how the genetic risk modulates this progression. This could be done by testing the interactions between a GDM/T2D PRS and T2D status in women with prior GDM. This could reveal whether women with prior GDM and a high genetic risk are more likely to develop T2D than women with prior GDM and a low genetic risk. Finally, given the low sensitivity of the PRS themselves, future studies should focus on deriving and estimating the predictive value of a composite score which combines the GDM/T2D PRS, family history of diabetes, prior GDM status, and diet quality score in order to improve the identification of women at higher risk of developing T2D.”

2) Are the 'Born in Bradford' participants in this study also in the Pervjakova paper? – Pervjakova et al. 2022, PMID: 35220425 which is referenced in the manuscript (as a medRxiv – now published). Please make clear what the degree of overlap between BIB participants is between these papers if any. Please be more explicit about what this paper adds to the Pervjakova analysis in the Discussion. This will be useful to readers, because that previous paper reached similar conclusions about the overlap between T2D and GDM genetic susceptibilities, and most of the statistical power here comes from the BIB cohort.

This is indeed an important point, which we now discuss on page 10 line 142 of the revised.docx manuscript (line 143 in.pdf). In brief: We mention the overlap between the BiB samples in the Pervjakova paper and ours, but state that it probably didn’t have a major impact given that BiB’s South Asian participants represent ~1.2% of Pervjakova’s total sample size. As for the added value of our manuscript, we state that the two studies reached a similar conclusion, but by using two different approaches (PRS in our case, locus by locus in Pervjakova) which helps strengthens the claim of a common genetic background.

3) The support for the hypothesis from the newly reported START cohort is strong and important. It seems unlikely that the interaction analyses would have had adequate power for the negative result to be interpreted as a strong negative. Can the authors comment on this, and possibly quantify the power they had?

Lack of power is indeed an issue, especially for interaction tests. We now acknowledge this in the discussion page 11 line 153 (line 154 in.pdf) by stating that:

“Most interactions tested were not significant in both studies. This absence of significance should however be treated with caution since our power analysis suggests that, given our sample size, we are only able to detect strong interaction effects for the majority of our tests”.

We also show the power estimations for the two strongest interactions (FPG ~ PRS x Diet and GDM ~ PRS x BMI) in the Results section page 19 lines 117 and 123 (lines 119 and 125 in.pdf version). Although we did calculate it, we do not show power tests results for the FPG ~ PRS x BMI and FPG ~ PRS x Birth in South Asia (y/n) since the interaction are significant (or close to significance) in both studies but have opposite direction of effect, which, as mentioned on page 11 line 158 (159 in the.pdf) “precludes a power issue, and suggests differences in the effect of these environmental factors between the two studies, or possibly false positive results.”.

4) There is a debate about the utility of PRS in screening which goes beyond cost (p16); as the authors will know Wald and others have argued that for most diseases PRS, even with a high aetiological fraction, would lack sensitivity and specificity if used for screening. Readers might be interested to know what sort of sensitivity and specificity for GDM might result with the OR of 1.47 found here. The authors do use the phrase "enhanced predictive value" on p16 so I believe asking for that to be quantified is fair.

Thank you to the reviewer for this comment. We now estimate the sensitivity and specificity of the PRS as suggested on page 8 line 101 (102 in.pdf file), and discuss the point by stating that “despite a strong association, the PRS has a low discriminatory value (detection rate of 10% for a 5% false positive rate) regarding GDM cases. This is in line with the observations of Wald et al. stating that most polygenic scores of complex traits derived to date would perform poorly as a screening tests in a clinical setting (Wald and Old, 2019).” (Page 12, line 178 of the revised.docx manuscript , and line 179 in the.pdf file).